# Defining a “Good Death” in Pediatric Oncology: A Mixed Methods Study of Healthcare Providers

**DOI:** 10.3390/children7080086

**Published:** 2020-07-31

**Authors:** Mallory R. Taylor, Krysta S. Barton, Jenny M. Kingsley, Julia Heunis, Abby R. Rosenberg

**Affiliations:** 1Department of Pediatrics, Division of Hematology/Oncology, University of Washington School of Medicine, Seattle, WA 98105, USA; molly.taylor@seattlechildrens.org; 2Palliative Care and Resilience Lab, Center for Clinical and Translational Research, Seattle Children’s Research Institute, Seattle, WA 98101, USA; krysta.barton@seattlechildrens.org (K.S.B.); jenny.kingsley@seattlechildrens.org (J.M.K.); 3Cambia Palliative Care Center of Excellence, University of Washington, Seattle, WA 98195, USA; 4Treuman Katz Center for Bioethics, Center for Clinical and Translational Research, Seattle Children’s Research Institute, Seattle, WA 98101, USA; 5Department of Pediatrics, Division of Critical Care Medicine, University of Washington School of Medicine, Seattle, WA 98105, USA; 6School of Medicine, University of California San Francisco, San Francisco, CA 94143, USA; julia.heunis@ucsf.edu; 7Department of Pediatrics, Division of Bioethics/Palliative Care, University of Washington School of Medicine, Seattle, WA 98105, USA

**Keywords:** adolescent and young adult, palliative care, qualitative, cancer, interdisciplinary

## Abstract

Delivering optimal end-of-life (EOL) care to children and adolescents is a healthcare priority, yet relatively little is known about what patients, families, and healthcare providers (HCPs) consider “best” practices. The objective of this study was to identify factors that pediatric oncology HCPs consider important for EOL care. This was a cross-sectional mixed methods study. Participants were multidisciplinary pediatric oncology staff who completed surveys and participated in semi-structured qualitative interviews. Interviews were analyzed using a modified grounded theory approach. Provider statements were compared based on years of experience (≤10 or >10 years) and discipline (non-physician or physician). A total of *n* = 19 staff (74% female) enrolled, including physicians (*n* = 8), advanced practice providers (*n* = 4), nurses (*n* = 2), music/art therapists (*n* = 2), physical therapists (*n* = 1), educators (*n* = 1), and chaplains (*n* = 1). Most HCPs identified communication, symptom control, and acceptance as features of a “good” death. Compared to physicians, non-physicians focused on relationships (67% vs. 33%, *p* = 0.007); HCPs with ≤10 years of experience (*n* = 11) more frequently identified the benefits of a multidisciplinary team (74% vs. 26%, *p* = 0.004). This study identified many common HCP-defined components of “good” pediatric EOL care in addition to some differing perspectives depending on discipline and experience. Incorporating diverse HCP perspectives with those of the patient and family can guide contemporary high-quality pediatric EOL clinical care and education.

## 1. Introduction

Empirical descriptions of a “good death” exist for older adults, and these have served as the foundation for providing high-quality end of life (EOL) and palliative care for these patients [1,2,3]. Medical institutions such as the World Health Organization, Institutes of Medicine and the American Society of Clinical Oncology recommend the integration of evidence-based palliative care practices into standard clinical care for patients [4,5,6]. Research focused on patient, family, and provider experiences have informed these recommendations in adults, leading to the development and dissemination of national clinical care guidelines [6,7].

There have been comparatively fewer studies in pediatric populations. Thus, less is known about what, if anything, constitutes a “good” death from the perspective of children and adolescents and young adults (AYAs), their caregivers, or their medical teams [8,9,10,11,12,13]. This gap is due, in part, to the inherent challenges of conducting rigorous research in this field, including smaller sample sizes, complexities of proxy reporting, and ethical concerns about studying this vulnerable population [14]. As a result, the current state of pediatric palliative care (PPC) delivery may not meet the EOL needs of patients and families [15,16,17].

The inclusion of healthcare provider (HCP) voices can contribute to improved PPC care. Pediatric oncology providers care for many patients at the EOL and can offer a breadth of insight regarding these difficult situations. Incorporating provider experience can identify common pitfalls, as well as successful strategies, to inform PPC delivery. HCP experience can also serve as a foundation for high quality clinical education, as there are currently no evidence-based standards in PPC training. Although the Accreditation Council for Graduate Medical Education (ACGME) requires Pediatric Hematology/Oncology fellows to “integrate palliative care for patients with hematologic and oncologic conditions”, there are no guidelines for training, nor are PPC concepts part of board certification exams [18,19]. The objective of this study was to elicit pediatric oncology provider perspectives on factors that contribute to a positive dying experience for children, AYAs, and their families.

## 2. Materials and Methods

This was a cross-sectional mixed methods study. Eligible participants were any healthcare provider (HCP) of pediatric and AYA patients with cancer treated at Seattle Children’s Hospital. Recruitment followed a snowball approach: first, the AYA Psychosocial Team identified and contacted HCPs known to be involved in the care of patients with cancer who had died, and then additional participants were recruited through peer referral. The study was presented to participants as an effort to understand HCP perspectives on what is important for AYA patients and families at the end of life. Following enrollment, participants filled out novel online surveys querying demographic information, clinical experience (including years in practice and average monthly patient deaths), as well as formal training in, and self-identified comfort with, EOL communication.

For the qualitative component of the study, trained research staff (a pediatric oncology/palliative care physician who interacted professionally with the staff (A.R.R.)) and two research associates with no professional interactions with staff (non-authors) conducted in-person semi-structured interviews between November 2014 and April 2015. Enrolled HCPs were asked about patient deaths that, from their perspective, had gone well or poorly (“Have you ever seen or heard of someone dying in a way you thought was particularly good or bad?”) and factors they felt had contributed to each outcome. Individual interviews took place at the hospital at a mutually agreed upon time.

We followed the Standards for Reporting Qualitative Research (SRQR) guidelines for the conduct of qualitative research studies [20]. Data were analyzed using ATLAS.ti software (ATLAS.ti Scientific Software Development GmbH, Berlin, Germany). We employed a modified grounded theory approach and applied a constructivist paradigm to data collection and analysis in order to gain perspective about the subjects’ own environment [21]. The study interview guide (see Appendix A) was iteratively adapted based on emerging themes and participant feedback and consisted of open-ended questions with probes [22]. Our initial target sample size of *n* = 15 was extended due to additional HCPs expressing interest in participation and our goal of reaching theoretical saturation [22]. Each interview was audio recorded verbatim, de-identified, and transcribed. Two primary investigators (M.T., a pediatric oncologist who interacted professionally with staff, and K.B., a senior health services researcher with no professional interactions with staff) independently read selected transcripts and utilized open coding techniques to develop a code book based on prominent themes and subthemes that were used as coding categories. The resulting codebook was iteratively refined until there were no further modifications needed when applied to subsequent transcripts.

Focused coding then began, with the primary coder (M.T.) and secondary coder (K.B.) evaluating each transcript sequentially. During the coding phase, a “constant comparative” technique was used to identify and validate themes related to a perceived “good” death in AYAs and children with cancer [21,22]. To enhance trustworthiness, investigators met regularly in person to review completed transcripts for validity and reach consensus. Where coders identified discrepancies, a third coder (J.K., a pediatric intensive care/bioethics physician with no regular professional interaction with staff) analyzed the relevant text to help reach consensus. Axial coding was then undertaken, examining the relationships between prominent themes (i.e., factors co-occurring with a “good” vs. “bad” death). To describe predominant themes endorsed by the full cohort of participants, we tallied the total number of quotes in each coding category under the theme and reported the top quartile.

For the mixed methods analysis [23], we sought to explore whether discipline and experience impacted the HCPs’ perspectives on EOL care, using both demographic survey data and qualitative data. We dichotomized HCPs by type of training (physician or non-physician) and years of experience (≤10 or >10 years) and compared the proportion of quotes in each theme. We totaled the number of quotes made by each HCP group for each theme and the percentage of statements by HCP group were calculated (e.g., 72% of all statements about “Relationships” came from non-physicians). To isolate the prominent, disparate perspectives, we set cut points for codes with at least 15 quotes and at least a 20% difference in prevalence of quotes between comparison groups. After looking at the frequency and distribution of the data, these cut points were chosen by the study team to focus on meaningful differences between groups, as these thresholds applied to approximately one quarter of quotes. Differences between types of statements based on HCP training and years of experience were calculated using Pearson χ^2^ analysis. For this exploratory analysis, we report all alpha levels and considered those <0.10 relevant for future hypotheses. All subjects gave their informed consent for inclusion before they participated in the study. The study was conducted in accordance with the Declaration of Helsinki, and the study protocol was approved by the institutional review board of Seattle Children’s Hospital (approval number FWA00002443).

## 3. Results

### 3.1. Demographic Characteristics

A total of *n* = 19 oncology medical staff members (74% female) participated (Table 1). Staff included physicians (*n* = 8), advanced practice providers (*n* = 4), and other multidisciplinary team members, such as physical therapists (*n* = 1), bedside nurses (*n* = 2), music/art therapists (*n* = 2), educators (*n* = 1), and chaplains (*n* = 1). Just over half (53%) self-reported receiving any training in communication skills around death and dying. Sixty-three percent self-reported receiving training to improve their ability to provide EOL/hospice or palliative care. When asked to rank their comfort level (0 = least comfortable, 10 = most comfortable) in having conversations with pediatric/AYA patients and conversations with family members, HCPs on average reported scores of 6.8 (range 2–10) and 7.2 (range 3–10), respectively.

### 3.2. Qualitative Analysis

Qualitative analysis of HCP interviews revealed ninety-five coding categories organized into twenty-six major themes (Appendix B, Table A1). The most frequently mentioned themes (top quartile across all HCP participants) included “Communication”, “Symptom Management”, “Acceptance”, “Meeting Families Where They Are”, “Expectation/Anticipation of Death”, “Support/Community”, and “Control” (Table 2).

Communication: Communication was the most frequently mentioned theme among HCPs. Providers identified early and transparent communication with patients and families as a central element of a positive EOL experience. Although many HCPs felt the timing of this type of EOL communication should happen only when agreed upon by the patient/family, other providers felt it was important to “push” families a bit to have these conversations even when they might not feel ready. They acknowledged the challenges of initiating and guiding these conversations, and simultaneously identified a lack of open and honest communication as a key factor in negative EOL experiences. One provider said, “It was a hard thing to realize that if we had just found a way to break down that communication barrier. We could have saved the child the experience of a prolonged death”.

Symptom Management: HCPs expressed the need for adequate symptom control for their patients. One provider stated “Any pain that we could not control is a failure on our part to provide good quality and ethical care to our patients”.

Acceptance: HCP participants pointed to parental acceptance as important for a “good” death, and conversely noted a patient/family’s lack of acceptance often correlated with a negative EOL experience. HCPs also acknowledged their own lack of acceptance, with statements like, “I’m still trying to give up the fact that I can’t cure the people that have already died”.

Meeting Families Where They Are: The concept of eliciting and honoring patients’ and families’ preferences was identified as an essential practice among HCPs. They said that prioritizing patient/family preferences above their own was important, as one HCP stated, “What I think is great or not great doesn’t really matter because whatever happens to be right for the family is what’s right”.

Expectation/Anticipation of Death: HCPs noted that, when possible, being able to plan for death was helpful to families. Many felt it was the medical team’s role to help prepare families for this process, as they have more experience with death. One provider said, “Patients are as prepared as we make them”. Other HCPs highlighted the need for a more multidisciplinary team approach to provide this anticipatory guidance. Generally, an unexpected or sudden death was often perceived as a “bad” death by HCPs.

Support/Community: Many HCPs felt that social support—whether it came from family and friends, their external community, or more formal supports like hospital-based Social Work and Palliative Care teams—was critical to families when their child was dying. HCPs noted that adolescents primarily wanted their peer group around at the EOL and also recognized that their support network may have been smaller at the EOL than earlier in their illness experience. “They want their world to be smaller, not bigger”.

Control: Maintaining a sense of control was perceived as valuable for patients and families to have a positive EOL experience. One HCP, speaking from a patient perspective, said, “Maybe the common theme is control—I [the patient] have more pain control, I’m more awake, I’m less awake, I’m at home, I’m not at home. But the point is I’m in control of each of those factors rather than the doctors or the cancer”.

### 3.3. Mixed Methods Results

Physician vs. Non-Physician HCPs. Ten themes met our criteria for comparison between physicians and non-physicians (Table 3, Figure 1). For example, compared to physicians, non-physician HCPs focused on clinician experience with EOL care (70% vs. 30%, *p* < 0.001), relationships with patients (67% vs. 33%, *p* = 0.007), and patient/family response to the dying process (69% vs. 31%, *p* = 0.003). Non-physicians also more commonly stated it was important to “meet families where they are” (62% vs. 38%, *p* = 0.05).

Physician HCPs tended to reference the concept of fear or worry more often than non-physicians (73% vs. 27%, *p* = 0.06, Table 3, Figure 1), suggesting that children most often worried about their family and friends rather than themselves. Physicians also felt it was important for families to have clear expectations about the dying process (61% vs. 39%, *p* = 0.07), often suggesting it was the duty of the medical team to give guidance to the patient and family around these expectations. One physician stated, “I think it really falls on the provider to help the families and the patients be prepared”. In contrast, a non-physician HCP said, “I have certain things that I say to families like ‘nobody knows them better than you.’” Physicians more frequently mentioned a sense of failure when a child died from their cancer (71% vs. 29%, *p* = 0.15) and also referenced hope as important to pediatric EOL care (77% vs. 23%, *p* = 0.1), although these differences did not meet our threshold for statistical significance.

Earlier vs. Later Career HCPs. Whereas many HCPs underscored the importance of communication, those with ≤10 years of experience (*n* = 11) more frequently did so (66% vs. 34%, *p* < 0.001, Table 3, Figure 2). In addition, less experienced HCPs more commonly identified a multidisciplinary team as an important component of good pediatric EOL care (74% vs. 26%, *p* = 0.004). They also referenced participation in research as important (76% vs. 24%, *p* = 0.002), which they most commonly associated with deriving a sense of meaning from the dying process. HCPs with ≤10 years of practice also noted the phenomenon of protection or selflessness (from patients or caregivers) around the time of death (74% vs. 26%, *p* = 0.003). Specifically, they tended to observe that patients or caregivers were protecting each other from the pain of talking about death. These HCPs also more frequently mentioned the concept of family expectation/anticipation of death (62% vs. 38%, *p* = 0.07), fear or worry (86% vs. 14%, *p* = 0.002), and the metaphor of a “fight” against cancer (74% vs. 26%, *p* = 0.02).

## 4. Discussion

In this mixed methods study, we endeavored to evaluate multidisciplinary pediatric HCP perspectives of a “good death” in pediatric patients with cancer. Common themes emerged from provider interviews, including a focus on pain and symptom management and good communication. These shared elements should serve as a framework for high quality, contemporary PPC clinical practice. Additionally, some HCPs defined “good” pediatric EOL care differently depending on years of experience and type of training. Non-physicians more commonly emphasized relationships and “meeting families where they are,” whereas physicians tended to emphasize guidance around end of life expectations. HCPs with more than 10 years of experience more commonly considered a family’s sense of control, and those with less experience emphasized the potential benefits of a multidisciplinary team. Notably, a majority of the prevalent and disparate themes were weighted toward HCPs with less than 10 years of experience. Although there were some coding categories with higher numbers of quotes among more experienced HCPs, they did not meet our inclusion thresholds.

Some of these themes are concordant with prior studies in the adult literature, such as pain and symptom management, and preparation for death [1]. This reinforces the need for HCPs to diligently address symptom management at the EOL, regardless of patient age. It also highlights the importance of preparing patients and families, when possible, for what the dying process could look like. Although predicting the exact timing of death is difficult [24], there is an opportunity to describe common EOL symptoms and typical progression.

That we observed differences in predominant themes by discipline and years of experience is not surprising; different disciplinary training necessarily targets different aspects of patient care, and a clinician’s breadth of prior experience necessarily impacts their opinions. For example, non-physicians may have more training in psychosocial care, thus recognizing the value of relationships in the critical EOL period. In contrast, physicians’ focus on anticipatory guidance needs may reflect their training as pediatricians who strive to prepare families for their child’s illness or wellness experiences.

Additionally, when applying value judgements (“good” or “bad”) to the EOL experience, it is important to clarify from whose perspective. Most often, HCPs referenced a death as “good” or “bad” in terms of the perceived patient/family experience. However, there were a small number of explicit examples of HCPs contrasting that what was “good” for the family was not necessarily “good” for the provider team. For example, performing intensive resuscitation interventions that were unlikely to be successful was emotionally challenging for the HCP, but was what the family needed to feel like their child received “good” EOL care. Overall, the HCPs interviewed felt that a “good” death was however the patient/family defined it, but there were occasional circumstances where their own personal definition could be in conflict.

It should also be noted that many of the coding categories represent complex, multifaceted concepts when caring for young patients and families at the EOL. For example, “Hope” is represented as a single theme, with most HCPs referencing this term when referring to patients/families’ optimism for a cure. However, this construct in children/AYAs with cancer could also include a diversity of hope under the larger positive psychology frameworks of “agency” and “pathway” [25]. Patients and families can hold many types of hope simultaneously (hope for lack of pain, returning home, meeting school goals, finding spiritual comfort, etc.), and yet many HCPs are unaware of the breadth of their patients’ hopes beyond miracles and cures [26,27].

Over half (63%) of HCPs interviewed self-reported that they had received training in palliative care skills, which is higher than generally reported in the pediatric oncology field. In a prior survey of pediatric oncology providers, only 10% reported any formal training in palliative care in medical school, and over 90% of clinicians said they learned to care for dying children by “trial and error [28]”. Hence, our participants provided informed insight into important elements of EOL care. Primary palliative care skills are an important component of pediatric oncology practice, and general oncology HCPs should be able to provide this service to patients and families. The themes highlighted in this study could help inform the development and refinement of pragmatic HCP EOL training.

This study has several additional limitations. First, all providers practiced at the same large, well-resourced academic center, and their experience may not be applicable to smaller, community-based programs. Additionally, because we had a small, heterogeneous study population, we cannot identify discipline-specific themes confidently and this may further limit generalizability. There was likely a broad range of diverse experiences within the participant groups (physicians, non-physicians, ≤10 years’ experience, >10 years’ experience), thus limiting the pooled group results. However, comparing groups based on these important differences still provides helpful practice insights. This study was not powered to detect quantitative differences in HCP subgroup perspectives, and the methodology for selecting meaningful cut points was determined by our team. That we identified several differences between groups in this exploratory analysis suggests there may be important differences in perspective worth evaluating more carefully. Likewise, we evaluated several possible associations and did not adjust our findings for multiple comparisons, raising the possibility of some false positive results. However, this exploratory analysis identifies relevant hypotheses for future study and is one of the largest mixed methods studies of pediatric oncology providers’ experiences in caring for children/AYAs at the end of their life.

## 5. Conclusions

In this mixed methods study of HCP providers of children and AYAs at the EOL, we identified several common themes describing “good” EOL care, including open communication and expert pain and symptom management. These shared observations represent key focus areas for clinicians caring for children at the end of life. In addition, we found important differences in HCP perspectives; discipline and years of experience may influence HCP definitions of quality EOL care. Incorporating these providers’ voices and experiences is an important step in delivering high-quality, standardized pediatric palliative care.

## Figures and Tables

**Figure 1 children-07-00086-f001:**
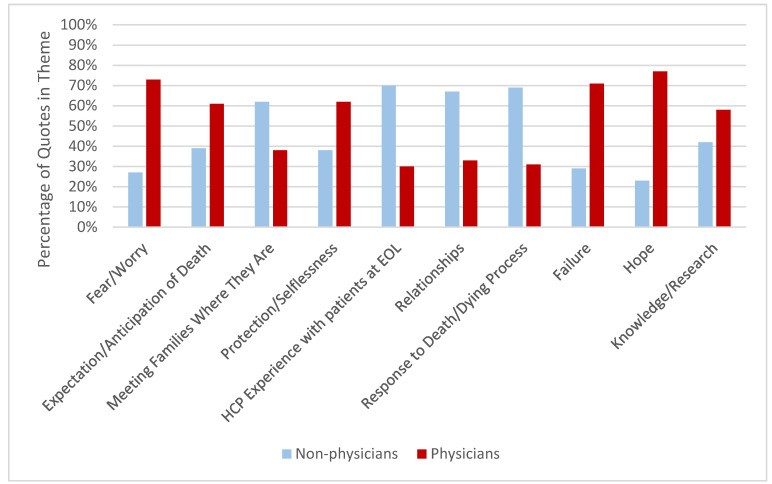
Prevalence of Quotes: Non-physician vs. Physician Healthcare Providers. Comparing non-physicians to physicians: prevalent and disparate themes.

**Figure 2 children-07-00086-f002:**
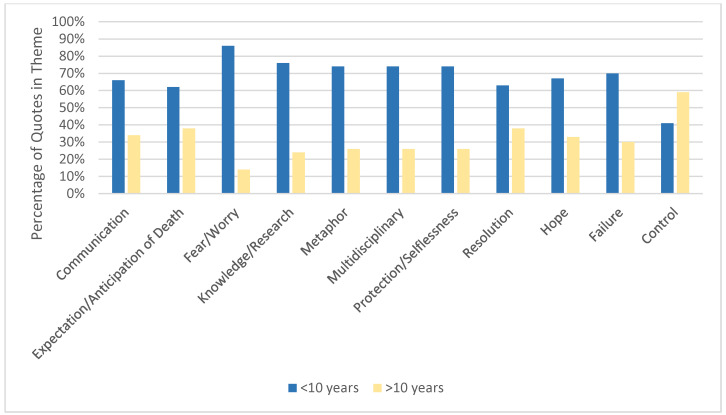
Prevalence of Quotes: Healthcare Providers with <10 years vs. >10 years experience. Comparing providers with differing amounts of practice experience: prevalent and disparate themes.

**Table 1 children-07-00086-t001:** Healthcare Provider Characteristics.

	Total = 19
	*n* (%)
**Role**	
Pediatric Oncology Physician/Fellow	8 (42)
Pediatric Oncology Advanced Practice Provider (NP/PA)	4 (21)
Pediatric Oncology Nurse	2 (11)
Other (Child Life Specialist, Chaplain, Physical Therapist, Educator)	5 (26)
**Sex**	
Male	5 (26)
Female	14 (74)
**Race/Ethnicity**	
Hispanic/Latino	0 (0)
American Indian/Alaska Native	1 (5)
Asian	3 (16)
Black	1 (5)
White	15 (79)
**Experience in End of Life Care**	
Number of Years in Position (median, range)	8 (<1–18)
Number of AYA/pediatric patient deaths per year	
*1–5*	14 (74)
*6–10*	3 (16)
*11–15*	2 (10)
Number of AYA/pediatric patients cared for per month (mean, range)	20 (5–75)
**Received training in: ^a^**	
*Communication*	
Yes	10 (53)
No	9 (47)
*Talking about death and dying*	
Yes	10 (53)
No	9 (47)
*Providing end of life/hospice/palliative care*	
Yes	12 (63)
No	7 (37)
**Comfort level in discussing death/dying with: ^b^**	Mean (range)
Patients	6.8 (2–10)
Parents/caregivers	7.2 (3–10)

Abbreviations: NP = Nurse Practitioner; PA = Physician’s Assistant. ^a^ Provider reported ever receiving any formal training. ^b^ Self-reported comfort with having discussions with patients or parents/caregivers, where 10 = most comfortable, 0 = least comfortable.

**Table 2 children-07-00086-t002:** Most Frequently Mentioned Themes Across All HCP Groups.

Theme *	Definition	Exemplary Quotes
Communication	*Interchange of information regarding prognosis, options, and preferences*	“It was a hard thing to realize that if we had just found a way to break down that communication barrier between the parents and the child that we could have saved the child the experience of a prolonged death”.
Symptom Management	*Means of minimizing mental and physical distress around death*	“First of all, a good death needs to be a comfortable death. The symptom piece of it needs to be managed”.
Acceptance	*Recognition and internalization of death as an inevitable outcome*	“The cultural shift now is that we’re just a little bit more accepting, and sometimes no matter what we do... we can’t save everybody”.
Meeting Families Where They Are	*Eliciting and accommodating patient/family preferences*	“What you need to do more than anything is listen to the cues of your families it’s meeting the family in the process, where they’re at”.
Expectation/Anticipation of Death	*Predicting and planning for timing and manner of death*	“I think the truth is patients are as prepared as we make them”.
Support/Community	*Social network available to patient/family*	“For adolescents the peer support group is more important than their parents in terms of how they see the world and who they see themselves to be”.
Control	*Exercising direction over aspects surrounding the dying process*	“I think [it’s important] that the family have the death go in the way that they had envisioned it”.

* Abbreviations: HCP = Healthcare Provider. * Total number of quotes were counted for each theme; the top quartile are presented here.

**Table 3 children-07-00086-t003:** Prevalence of Quotes Based on HCP Training and Experience ^a^.

	Count (%)	Count (%)	*p*-Value ^b^
Theme *	Non-Physician (*n* = 11)	Physician (*n* = 8)	
Expectation/Anticipation of Death	27 (39)	43 (61)	0.07
Fear/Worry	6 (27)	16 (73)	0.06
Meeting Families Where They Are	43 (62)	26 (38)	0.05
Protection/Selflessness	16 (38)	26 (62)	0.16
HCP Experience Working With Patients at EOL	85 (70)	37 (30)	<0.001
Relationships	68 (67)	33 (33)	0.007
Response to Death and the Dying Process	43 (69)	19 (31)	0.003
Hope	3 (23)	10 (77)	0.1
Failure	5 (29)	12 (71)	0.15
Knowledge/Research	16 (42)	22 (58)	0.42
	<10 years (*n* = 11)	>10 years (*n* = 8)	
Communication	98 (66)	51 (34)	<0.001
Expectation/Anticipation of Death	42 (62)	26 (38)	0.07
Fear/Worry	18 (86)	3 (14)	0.002
Metaphor	20 (74)	7 (26)	0.02
Multidisciplinary	29 (74)	10 (26)	0.004
Protection/Selflessness	31 (74)	11 (26)	0.003
Resolution	20 (63)	12 (38)	0.2
Hope	8 (67)	4 (33)	0.4
Failure	14 (70)	6 (30)	0.1
Knowledge/Research	29 (76)	9 (24)	0.002
Control	24 (41)	35 (59)	0.2

* Abbreviations: HCP = Healthcare Provider; EOL = End of Life. ^a^ HCPs were divided up by type of training (physician or non-physician) and years of experience (<10 or >10 years). All themes with at least 15 quotes by one group and at least a 20% difference, or those determined to be clinically relevant by the coding team, are presented here. ^b^ Pearson’s chi-squared analysis; *p*-value < 0.10 considered relevant for future hypotheses. * Theme Definitions: *Expectation/Anticipation of Death*: Predicting and planning for timing and manner of death. *Fear/Worry*: Expressing the emotion of fear regarding care or outcome. *Meeting Families Where They Are*: Eliciting and accommodating patient/family preferences. *Protection/Selflessness*: Concern for the wellbeing of others; trying to prevent suffering or harm. *HCP Experience Working with Patients at EOL*: Experience caring for cancer patients especially around the time of death. *Relationships*: Interpersonal dynamic between providers and patients/families. *Response to Death/Dying Process*: Description of coping strategies, emotions, etc. *Hope*: Faith or belief in the possibility of a good outcome or cure. *Failure*: Concept that death in pediatric cancer is always a bad outcome. *Knowledge/Research*: Expressing need to learn more about providing good end of life care. *Communication:* Interchange of information regarding prognosis, options, and preferences. *Metaphor*: Concept of a fight against cancer; conflict, war, battle. *Multidisciplinary:* Recognizing multiple members of the team and the importance of having a varied team. *Resolution:* Some form of closure or completion. *Control:* Exercising direction over aspects surrounding the process of dying.

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
