# Peer review of "Defining a “Good Death” in Pediatric Oncology: A Mixed Methods Study of Healthcare Providers"

_children, 2020, doi:10.3390/children7080086_

Round 1

Reviewer 1 Report

The authors correctly indicate that little is known about how health care professionals (HCPs) caring for children with cancer think about what constitutes a "good death." The manuscript reports on a mixed-method (survey + qualitative interviews) study of pediatric oncology HCPs regarding their views on end of life care.  The authors report their data based on the professional identity (physician or not) and experience (greater than or less than ten years) of the study participants, indicating commonalities and differences among the groups. 

General Comments: While perhaps not technically inadequate, the statistical aspects of the study seem problematic. The number of participants overall is small and among the non-physicians the professional roles are quite heterogeneous.  One wonders if it makes sense to imply generalizability given this.  Regarding "themes" that the authors report, it seems they really identify categories of topics discussed by the participants, rather than content-full themes.  The reader would learn substantially more if the authors told us more about the variability reported on such topics as fear, hope, protection, etc. Also, the authors should provide more information on how they presented the study to the participants.  Did they tell the subjects the research aimed to discover views about good deaths and, if so, how did they define that term? In other words, did the authors inadvertently bias the responses of their participants?

Specific comments:

Page 2, line 46: are the references correct?  From the context it seems 6 and 7 make more sense than 7 and 8.

Line 69: It would be best if the authors clarified that they mean oncology patients who have died (or not).

Page 3, line 104: as noted above, it does not seem appropriate to lump all non-physicians into a one "discipline."  The experiences of the a chaplain are different from those of a physical therapist.  These differences undermine the quantitative reporting.

Line 111: what was the basis for choosing a 20% difference in prevalence as a threshold value?

Line 121:  how did the authors define "formal training" for their participants?  A single didactic session about how to communicate with parents of a dying child is quite different from a series of role-playing sessions with formal feedback.

Lines 124-126: questions with self-report on comfort with clinical skill are notorious for lacking strong correlation with more objective assessments of performance.

Page 5, Table 2: The authors might want to expand on the implications of the quote noting that "sometimes...we can't save everybody."  The reality is that ~25% of pediatric cancer patients die of their disease.  Does "sometimes" suggest a misconception of this fact and thus bias views of end of life care?

Lines 154-160: [As an example] No doubt communication was a "theme."  What more can the authors tell us?  One guesses participants expressed a variety of views about what constitutes good communication for end of life.

Page 6, lines 189-194:  Again, is it valid to combine all the non-physicians?

Page 8. line 207: As Feudtner and others have delineated, "hope" needs to be unpacked. It seems the authors assume they know what participants meant.

Page 9: The authors should discuss how pediatric oncology HCPs should (or should not) distinguish their roles and practice from those of pediatric palliative care professionals.

Reviewer 2 Report

Comments:

I don’t have major concerns at all, though I have several thoughts which I have included below.  With some minor adjustments this paper will add important guidance to HCPs participating in EOL care of Pediatric Haem/Onc patients and their families.  

-In the abstract, the conclusion as stated, seems to detract from the thrust of the paper, as it highlights the differences in HCP’s rather than what exactly the elements of what makes “good death”.

-In the introduction, I was surprised by the focus on improving education, as there is no hint of this as being a study meant to inform educational standards in the title and abstract.  That being said, it is a great reason to do this work! I think if it is a main desired outcome, then it could be inserted as a goal of the study in the abstract.

The main point the authors seem to be  driving at is: ‘Understanding what HCPs feel constitute a “good death” will contribute to improved PPC care.’  But in some ways this message is diluted in the introduction, by spending considerable time discussing barriers related to the population itself (not HCPs), and educational short falls.  

-I wondered whether instead of only years of experience and type of training, the statistics could have included number of children they cared of that died.  There are certainly pockets of Paeds Haem/Onc that see more death than others, so while experience is a worthy surrogate in most cases, I found myself wondering whether actual experience (“reps”) would have had a large effect on finding as well.  The data is already recorded, so it should be able to be done?

-In the results, I really liked the quotes added. I especially found it interesting that one of the themes was “acceptance”. Not as usually thought of with regards to patients/families, but rather the HCPs themselves! This is such an important message, as I am quite sure that many HCPs consideration that a death is “bad” is a reflection of their own demons as much as anything else.  This is also reflected in the bar graph which shows that failure is much more prevalent in the MD group.  

-Also the pressure HCPs place on themselves shone through, with the quote that we any pain not controlled is failure

-Very interesting that the HCP>10yrs loom like they just said less about everything as depicted in the graph about ‘quotes in theme’.  I wonder if the authors could expand on why they think that it, and what it means?

-In the discussion, I will reiterate that, while interesting, the focus on the differences in care providers in some ways detracts from the excellent work of identifying these critical elements of a “good death”, that the authors have attained in their study. I understand that they call for future research to ensure a variety of HCPs and experience levels, however, I think the tone of the discussion should err further on the side of making clear the areas that HCPs can focus on in helping families/children attain the best dying/death/bereavement experience possible  

Finally, I think that the authors must be very clear who the HCP’s are referring to in saying it was a good or bad death.  In other words, I think it is a very different thing to say, that “it was a good death for the patient/family”, than “I felt as a HCP it was a good death (or as good as we could manage)”  these are both important things, but the latter is more introspective, and personal on the part of the HCP. While the former is an interpretation of an external source of information 

Round 2

Reviewer 1 Report

The authors have responded appropriately to reviewer comments.  The word "key" is overused and the number of times it is used should be reduced. 

Author Response

Thank you for noting the overuse of this word, and we agree.  We have eliminated all but two of the uses of the word "key."